# Impact of the Severity of Acquired von Willebrand Syndrome on the Short-Term Prognosis in Patients with Temporary Mechanical Circulatory Support

**DOI:** 10.3390/medicina58020238

**Published:** 2022-02-04

**Authors:** Makiko Nakamura, Teruhiko Imamura, Hiroshi Ueno, Koichiro Kinugawa

**Affiliations:** Second Department of Internal Medicine, University of Toyama, Toyama 930-0194, Japan; nakamura@med.u-toyama.ac.jp (M.N.); hueno@med.u-toyama.ac.jp (H.U.); kinugawa@med.u-toyama.ac.jp (K.K.)

**Keywords:** ventricular assist device, extracorporeal membrane oxygenation, Impella, shear stress

## Abstract

*Background and Objectives:* Acquired von Willebrand syndrome (AVWS) develops not only in patients with durable ventricular assist devices but also in patients receiving temporary mechanical circulatory support (MCS). However, its prognostic implication remains unknown. *Materials and Methods:* Patients who received temporary MCS in our institute between August 2018 and September 2021 were included in this prospective study and the von Willebrand factor multimer analyses were performed following the initiation of temporary MCS supports. The von Willebrand factor large-multimer index was calculated as a normalized ratio of large-multimer proportion among total von Willebrand factor. Association between the large-multimer index and the 30-day survival was investigated. *Results:* A total of 31 patients (69 years old, 52% men) were included. Median large-multimer index was 63.0% (56.9%, 75.6%). The index was lowest in patients with extracorporeal membrane oxygenation than those receiving support from other devices. A lower index (<59.9%) was associated with lower 30-day survival (41.7% versus 94.7%, *p* = 0.001) with an odds ratio 0.044 (95% confidence interval 0.002–0.805, *p* = 0.035) adjusted for other potential confounders. *Conclusions:* An advanced AVWS was associated with lower short-term survival in patients with temporary MCS. The clinical implication of AVWS-guided temporary MCS management remains the next concern.

## 1. Introduction

The use of temporary mechanical circulatory support (MCS) devices including trans-catheter left ventricular assist device (LVAD) Impella, extracorporeal membrane oxygenation (ECMO), and intra-aortic balloon pump, is increasing in patients with cardiogenic shock as a bridge to recovery, decision, and more intensified therapies [1]. However, bleeding remains one of the unsolved and often fatal MCS-related complications [2].

Bleeding is one of the major complications during MCS therapy, occurring in about 20–25% of patients and is associated with morbidity and mortality [2,3]. One of the potential causes of MCS-related bleeding is acquired von Willebrand syndrome (AVWS), which is caused by the loss of high-molecular-weight multimers of von Willebrand factor (vWF) [4]. Its detailed mechanism requires further study, but current etiological models for AVWS are an increased cleavage by the metalloprotease ADAMSTS13, mechanical destruction of vWF, and shear-induced vWF binding to platelets [4].

AVWS was originally observed in patients with durable LVADs. Recently, AVWS was found to develop also during other MCS supports using Impella and ECMO [5]. AVWS seems to develop within the initial hours and persists during the whole period of MCS therapy, often accompanying thrombocytopenia and bleeding events [2,4,6]. However, the impact of AVWS on mortality during these temporary MCS remains unknown. In this study, we investigated the severity of AVWS during temporary MCS and the impact of AVWS on the 30-day survival.

## 2. Methods

### 2.1. Patient Selection

Patients who received Impella (2.5, CP, or 5.0) and/or veno-arterial ECMO supports at our institute to treat cardiogenic shock between March 2018 and September 2021 were considered for inclusion in this prospective study. They received von Willebrand factor (vWF) large-multimer analyses at hemodynamically stable condition following the initiation of MCS supports. The present study was approved by the local institutional ethical committee and all patients gave informed consents before the initiation of MCS supports.

### 2.2. Device Selection

Impella type was determined according to the diameter of access vessels, required device flow, and support duration. Veno-arterial ECMO was used concomitantly with Impella or intra-aortic balloon pump, if necessary. All these decisions were made by the multidisciplinary team discussion.

### 2.3. Clinical Management

All patients received anticoagulation therapy with heparin purge solution and/or systemic intravenous heparin administration, targeting activated whole clotting time ranging from 160–180 s and activated partial prothrombin time ranging from 50–70 s.

If the hemoglobin level decreased below 10 g/dL, the red blood cell transfusion was considered. If the platelet count decreased below 5.0 × 10^4^ /μL and the patient had refractory bleeding at access site, mucosal bleeding, or hemoptysis, the platelet transfusion was considered. In case of the activated whole clotting time and activated partial prothrombin time prolongation over the targeting range, the fresh, frozen plasma was considered to be transfused. 

### 2.4. Measurement of vWF Multimer Index

vWF large-multimer analysis was performed using blood samples obtained at hemodynamically stable condition following the initiation of MCS supports, according to the method proposed by Ruggeri and Zimmerman (SRL Co., Tokyo, Japan) [7]. vWF large multimers were quantified from images of vWF multimer analysis by gel electrophoresis (Figure 1A). Using ImageJ, the central one-third of the lane was scanned. vWF multimers were divided into three parts (large, medium, and small/smallest multimers). A large-multimer ratio was calculated as the ratio of the large-multimer area to the total-multimers area (Figure 1B). A vWF large-multimer index was defined as a large-multimer ratio normalized by the control one (Figure 1C) [8,9]. In summary, a lower vWF multimer index indicates more progressed AVWS.

### 2.5. Data Collection

Laboratory data obtained in the same day of vWF large-multimer analyses were collected. Transthoracic echocardiography was performed on admission. A primary endpoint was 30-day survival. Secondary endpoints were hemocompatibility-related adverse events, including gastrointestinal bleeding accompanying melena, epistaxis, and progressive anemia with fecal occult blood, stroke, and thromboembolism.

### 2.6. Statistical Assessments

Statistics were performed using JMP pro ver14.0 (SAS Institute Inc., Cary, NC, USA). Variables with *p* < 0.05 were considered significant. Continuous data were described as median and interquartile range and compared between two groups using the Mann–Whitney U test. Categorical data were compared between two groups by Chi-square test or Fischer’s exact test as appropriate. Association between vWF large-multimer index and other clinical parameters was investigated by Pearson’s correlation coefficient.

The primary endpoint was defined as 30-day survival and the secondary endpoint was defined as hemocompatibility-related adverse event. The impacts of clinical variables including vWF large-multimer index on the primary and secondary endpoints were investigated by the logistic regression analyses. The odds values are expressed per unit increase in continuous variables. Variables with *p* < 0.05 in the univariable analyses were included in the multivariable analyses. A cutoff of vWF large-multimer index for 30-day survival was calculated by the Youden method in the receiver operating characteristics analysis and the cohort was stratified into two groups: lower vWF multimer index group and not-lower group. The area under curve was assumed as C-statistic. Continuous variables were transformed into dichotomized variables using a cutoff calculated by the receiver operation characteristics analyses for the primary endpoint. The 30-day survivals were compared between the dichotomized groups of vWF multimer index by Kaplan–Meier analyses and the log-rank test. 

## 3. Results

### 3.1. Baseline Characteristics

Of all potentially eligible patients, 31 patients with temporary MCS (Impella (2.5, CP, and 5.0) and/or ECMO) were included (Table 1). The median age was 69 years old and 16 (52%) were men. Twenty patients (65%) had ischemic etiology and 6 patients (19%) had dilated cardiomyopathy. ECMO was used in 10 patients. Half of them received both ECMO and Impella (i.e., ECPELLA). The median left ventricular ejection fraction was 25%. 

### 3.2. vWF Large-Multimer Index

The blood samples were obtained at 4 (2, 8) days following the initiation of MCS supports on median. vWF large-multimer index ranged between 37.1% and 98.9% with median value of 63.0% (Figure 2).

vWF large-multimer index had a moderate negative correlation with lactate dehydrogenase levels (*r* = −0.436, *p* = 0.014). There was also a negative correlation between vWF large-multimer index and the amount of blood product transfusions during MCS support, although not statistically significant (total blood transfusion, *r* = −0.317, *p* = 0.083; red blood cell, *r* = −0.321, *p* = 0.078; fresh frozen plasma, *r* = −0.056, *p* = 0.767; platelets, *r* = −0.324, *p* = 0.076).

### 3.3. Summary of Clinical Outcomes

There were 27 cases (87%) of hemocompatibility-related adverse events during MCS support. Of them, 9 cases (29%) included gastrointestinal bleeding, 4 cases had strokes, 1 case had limb ischemia during MCS and other 13 cases included cannulation site bleedings and unknown origin bleedings. There was no apparent clinical pump thrombosis. 

Eight patients died during the 30-day observational period, with 74.2% of survival of the study population. Five patients died due to sepsis and the other three patients died due to multiple organ failure. Deceased patients had received significantly higher amount of blood transfusion compared with the 30-day-surviving patients (total blood transfusion, 57 units versus 6 units on median, *p* = 0.014; red blood cell, 23 units versus 6 units on median, *p* = 0.013; fresh frozen plasma, 3 units versus 0 units, *p* = 0.039; platelets, 25 units versus 0 units, *p* = 0.010).

### 3.4. Impact of vWF Large-Multimer Index on the Primary Endpoint

Impact of baseline characteristics including vWF large-multimer index on the primary endpoint was investigated (Table 2). A cutoff of vWF large-multimer index to predict the primary endpoint was calculated as 59.9% with an area under curve 0.804 (78.3% of sensitivity and 87.5% of specificity; Figure 3).

The unadjusted odds ratio of vWF large-multimer index < 59.9% for the primary endpoint was 0.044 (95% confidence interval 0.004–0.403, *p* = 0.0008), and the adjusted odds ratio using platelet <9.8 × 10^4^ /μL and ECMO use was 0.040 (95% confidence interval 0.002–0.805, *p* = 0.035). Both potential confounders also tended to be association with 30-day survival (48% versus 88%, *p* = 0.095, 17% versus 75%, *p* = 0.006, respectively). C-statistics of these three variables that were enrolled in the multivariable analysis were stated in Table 3. 

The 30-day survival was significantly lower in the low-vWF, large-multimer index (<59.9%) group compared with others (41.7% versus 94.7%, *p* = 0.001, Figure 4).

### 3.5. Impact of vWF Multimer Index on the Secondary Endpoint

Hemocompatibility-related adverse events were observed in 27 cases (87%). vWF multimer index was not associated with hemocompatibility-related adverse events (*p* = 0.349), whereas device support duration, serum creatinine level, and ECMO use were significantly associated with the events (*p* < 0.05 for all). 

vWF multimer index was not associated with gastrointestinal bleeding (*p* = 0.227), whereas device support duration was associated with the events (*p* = 0.0061). 

### 3.6. Characteristics of the Patients with Low vWF Multimer Index

The low-vWF large-multimer index (<59.9%) group was associated with the ECMO use (67% versus 11%, *p* = 0.0020; Table 4). The values of vWF large-multimer index in patients with ECMO support (i.e., ECMO and intra-aortic balloon pump or ECMO and Impella) were lower compared with those with other devices (i.e., Impella 2.5 alone, Immpella CP alone, and Impella 5.0 alone), although some of their differences did not reach statistical significance (Figure 5).

## 4. Discussion

In this study, we investigated the association between vWF large-multimer index and 30-day mortality as a primary concern. The low-vWF large-multimer index (<59.9%) measured during temporary MCS was independently associated with 30-day mortality but was not significantly associated with hemocompatibility-related adverse events.

### 4.1. vWF Multimer Index

vWF multimer index was originally proposed by Horiuchi et al. to quantify the loss of large vWF multimers, indicating the severity of hematological AVWS [8,9]. In a recent study, mean vWF multimer index was as low as 33.8% at 591 days following the initiation of durable LVAD. Those with gastrointestinal bleedings had lower vWF multimer index with the cut-off value of 40% [9]. In our study, median vWF multimer index was 63.0% on median at 4 days following the initiation of temporary MCS. We did not follow the index, but long-term temporary MCS might also decrease the index and worsen AVWS in the same manner with durable LVAD.

### 4.2. A Decrease in vWF Multimer Index

Several studies investigated the difference in the severity of AVWS in each type of durable LVAD [9], whereas there are few studies comparing the severity of AVWS between Impella and ECMO. Given our finding, the use of ECMO seems to be associated with more advanced AVWS than Impella.

Detailed mechanism remains unclear, but ECMO support seems to cause higher shear stress together with a blood–air interface, resulting in more advanced AVWS [10]. A relatively rough configuration of the ECMO cavity surface also seems to cause more advanced AVWS [11]. The median level of lactate dehydrogenase was consistently higher in the lower multimer index group.

### 4.3. Prognostic Impact of Low vWF Multimer Index

Among those with durable LVAD, vWF multimer index <40% was associated with the occurrence of gastrointestinal bleeding [9], whereas the index was not associated with hemocompatibility-related adverse events in this study. The index was measured at day 4 on median and the median index value was relatively higher than those of durable LVAD (i.e., 63.0%). In critically ill patients with end-organ dysfunction under temporary MCS support who have many large cannulas, rapid decrease of vWF large multimer would easily cause superficial hemorrhagic events. However, the threshold values of vWF large-multimer index under temporary MCS on the occurrence of gastrointestinal bleeding in such patients may differ from those under durable LVADs.

The causes of hemocompatibility-related adverse events are multifactorial. As also observed in this study, long-term device supports would rather be a dominant contributor of the events. Although we did not assess, as reported in the durable LVAD cohorts, angiopoietin-2-related inflammatory and angiogenesis systems might also have a considerable impact on the events [12].

Instead, low multimer index was independently associated with short-term mortality in this study. The causes of death were sepsis or multiple organ failure. In patients with cardiogenic shock accompanied by bleeding, the immunity might have been reduced and sepsis and multiple organ dysfunction syndrome might also have developed, similarly to the patients after traumatic injury [13], although blood product replacement therapy had been received as treatment therapy of AVWS.

### 4.4. Recommended Strategy to Manage AVWS

When indicated, prompt device weaning is the therapeutic option for recovery for AVWS [4].

We recommend the following strategies in patients with severe cardiogenic shock with end-organ dysfunction: (1) Use temporary MCS devices in combination to recover end-organ failure promptly, paying attention to the presence and progression of AVWS; (2) attempt early weaning of ECMO with Impella 5.0 or CP as soon as possible; (3) attempt early device removal or bridge to durable LVAD, which had lower shear stress and less hemocompatibility-related adverse events [14].

Target-controlled anticoagulation therapy based on vWF parameters and other coagulations is a further concern. The implication of these therapeutic strategies should be validated in the next prospective study.

### 4.5. Limitations

This study was conducted in a single center using a small-sized cohort. Given the small event number, it is challenging to enroll multiple variables in the multivariable analyses. We enrolled three variables in this study, but this had a potential of overfitting. We should exhibit caution in interpreting the findings of the multivariable analysis; we did not have complete data of vWF antigen, ristocetin cofactor, or factor VIII activity, nor any collagen binding activity. Therefore, we evaluated the severity of AVWS only by vWF large multimer analysis. We administered antithrombin gamma (genetic recombination) for the management of temporary MCS, but it was not included in the amount of blood product. We measured vWF multimer index only one time for each patient. The prognostic impact of change in this index remains unknown.

## 5. Conclusions

An advanced AVWS was not associated with hemocompatibility-related adverse events but was associated with 30-day mortality following the initiation of temporary MCS. The implication of AVWS-guided therapeutic strategy for those with temporary MCS remains a future concern.

## Figures and Tables

**Figure 1 medicina-58-00238-f001:**
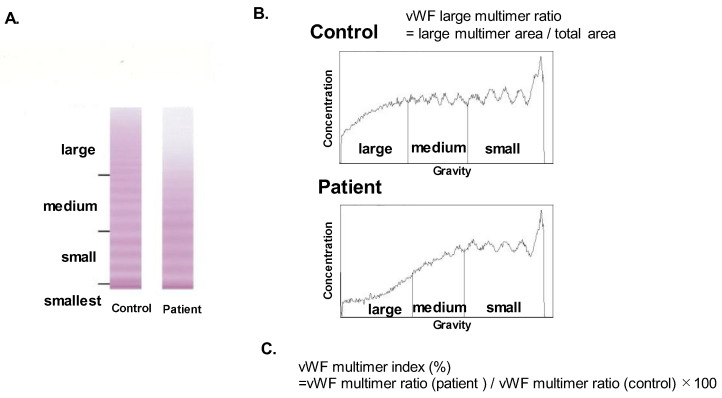
vWF multimer analysis by gel electrophoresis (**A**), the calculation of large-multimer ratio (**B**), and the calculation of large-multimer index (**C**). vWF multimers were divided into three parts (large, medium, and small and smallest multimers) according to their gravities (**A**). The large-multimer ratio was calculated as a ratio of the large-multimer area to the total multimers area (**B**). The large-multimer index was defined as a large-multimer ratio normalized by the value of control (**C**). vWF, von Willebrand factor.

**Figure 2 medicina-58-00238-f002:**
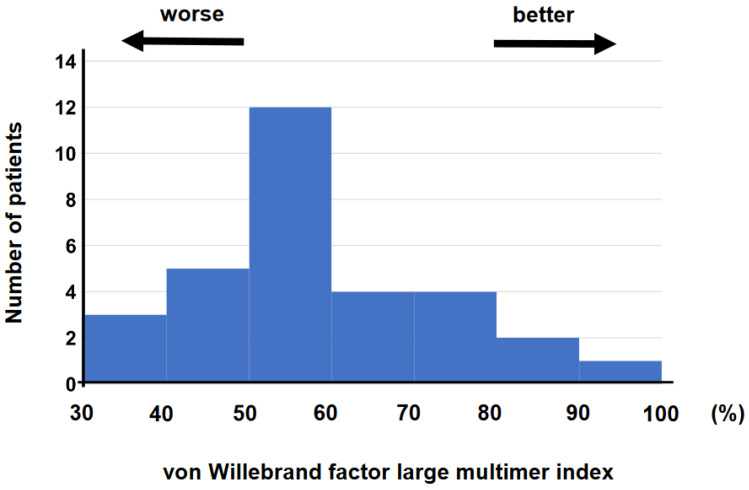
Histogram of vWF large-multimer index was shown. vWF—von Willebrand factor.

**Figure 3 medicina-58-00238-f003:**
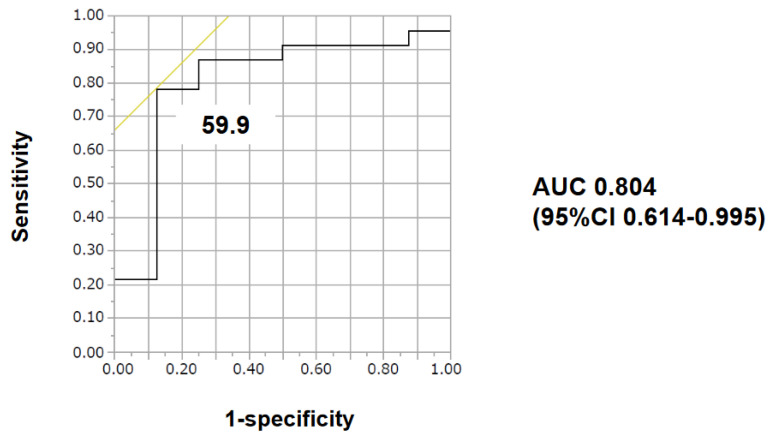
The cut-off of vWF large-multimer index for 30-day survival was calculated as 59.9 with an area under the curve of 0.80435 by receiver operating characteristics analyses. AUC—area under the curve; CI—confidence interval.

**Figure 4 medicina-58-00238-f004:**
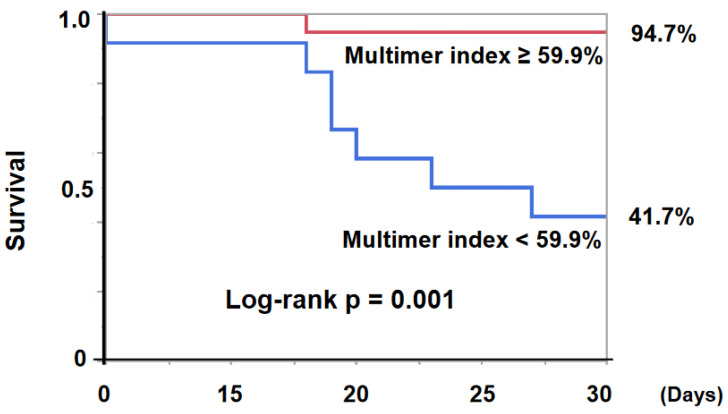
30-day survival was significantly lower in the low-vWF, large-multimer index (<59.9%) group compared with not low index (≥59.9%) group (41.7% vs 94.7%, log-rank *p* = 0.0010). vWF—von Willebrand factor.

**Figure 5 medicina-58-00238-f005:**
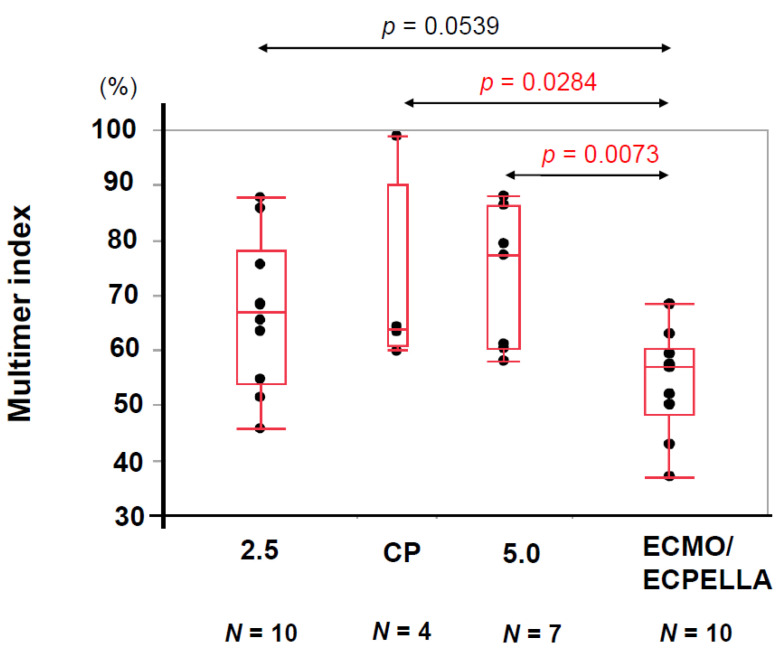
Comparison of vWF large-multimer index in each device were shown. The median values of vWF large-multimer index of Impella 2.5, Impella CP, Impella 5.0, and ECMO/ECPELLA were 66.9, 63.9, 77.3, and 57.1, respectively. The differences of vWF large-multimer index between Impella CP and ECMO/ECPELLA, and Impella 5.0 and ECMO/ECPELLA were 0.0284 and 0.0073 by Wilcoxon rank sum test. vWF, von Willebrand factor; ECMO, extracorporeal membrane oxygenation; ECPELLA, extracorporeal membrane oxygenation, and Impella.

**Table 1 medicina-58-00238-t001:** Baseline characteristics.

	*N* = 31
age (years)	69 (60, 82)
male sex	16 (52%)
ischemic etiology	20 (65%)
dilated cardiomyopathy	6 (19%)
others	5 (16%)
left ventricular ejection fraction (%)	25 (20, 36)
Impella 2.5	14 (45%)
Impella CP	5 (16%)
Impella 5.0	7 (23%)
extra corporeal membrane oxygenation	10 (32%)
extra corporeal membrane oxygenation and Impella	5 (16%)
systolic blood pressure (mmHg)	94 (79, 108)
mean arterial pressure (mmHg)	73 (63, 76)
pulse pressure (mmHg)	32 (8, 55)
heart rate (bpm)	90 (80, 106)
hemoglobin (g/dL)	10.2 (9.3, 11.1)
platelet (×10^4^/μL)	8.0 (5.9, 13.5)
total bilirubin (mg/dL)	1.4 (1.0, 2.2)
serum creatinine (mg/dL)	1.26 (0.83, 1.79)
C-reactive protein (mg/dL)	4.3 (1.8, 12.3)
B-type natriuretic peptide (pg/mL)	437 (186, 646)
log_10_ B-type natriuretic peptide (pg/mL)	2.64 (2.27, 2.81)
activated partial thromboplastin time (s)	50.1 (41.3, 65.5)
lactate dehydrogenase (U/mL)	838 (607, 1530)
Von Willebrand factor large-multimer index (%)	63.0 (56.9, 75.6)
antiplatelets	16 (52%)
Support duration (days)	4 (2, 8)

**Table 2 medicina-58-00238-t002:** Logistic regression analysis for the 30-day survival.

Univariable Analysis	*p*-Value	Odds Ratio	95% CI
age (years old)	0.991	1.000	0.943–1.060
male sex	0.076	4.667	0.766–28.406
mean arterial pressure (mmHg)	0.196	1.065	0.966–1.175
pulse pressure (mmHg)	0.352	1.017	0.981–1.054
left ventricular ejection fraction (%)	0.993	1.002	0.946–1.062
hemoglobin (g/dL)	0.203	1.511	0.780–2.927
platelet (×10^4^/μL)	0.024	1.278	0.979–1.669
platelet < 9.8 × 10^4^/μL	0.038	0.131	0.014–1.242
total bilirubin (mg/dL)	0.365	0.878	0.663–1.163
serum creatinine (mg/dL)	0.262	0.405	0.081-2.026
C-reactive protein (mg/dL)	0.272	0.948	0.864–1.042
log_10_B-type natriuretic peptide	0.436	1.939	0.359–10.460
activated partial thromboplastin time (s)	0.747	0.993	0.954–1.033
vWF large-multimer index (%)	0.0318	1.079	0.995–1.171
vWF large-multimer index < 59.9%	0.0008	0.040	0.004–0.403
support duration (days)	0.223	0.856	0.664–1.102
Impella 2.5	0.612	1.528	0.293–7.945
Impella CP	0.998	NA	NA
Impella 5.0	0.439	2.471	0.250–24.463
extracorporeal membrane oxygenation	0.007	0.070	0.011–0.483
ECPELLA	0.436	0.450	0.060–3.353
antiplatelets	0.477	0.550	0.106–2.860
multivariable analysis	*p*-value	Odds ratio	95% CI
platelet < 9.8 × 10^4^/μL	0.172	0.126	0.006–2.462
vWF large-multimer index < 59.9%	0.035	0.044	0.002–0.805
extracorporeal membrane oxygenation	0.691	0.572	0.036–9.024

ECPELLA—extracorporeal membrane oxygenation and Impella support; vWF—von Willebrand factor; CI—confidence interval.

**Table 3 medicina-58-00238-t003:** C-statistics of variables that were enrolled in the multivariable analysis.

	C-Statistics	95% CI
Platelet	0.747	0.545–0.950
vWF large-multimer index	0.804	0.614–0.995
extracorporeal membrane oxygenation	0.788	0.589–0.987

**Table 4 medicina-58-00238-t004:** Association between low vWF large-multimer index group and other clinical parameters.

	vWF Multimer Index (<59.9%) (*N* = 12)	vWF Multimer Index (≧59.9%) (*N* = 19)	*p* Value
age (years)	67.5 (57, 71.8)	71 (65, 83)	0.173
male sex	5 (42%)	11 (58%)	0.473
ischemic etiology	10 (83%)	10 (52%)	0.128
dilated cardiomyopathy	1 (8%)	5 (26%)	0.363
others	1 (8%)	4 (21%)	0.624
left ventricular ejection fraction (%)	25 (21, 48.5)	30 (20, 26)	0.951
Impella 2.5	6 (50%)	8 (42%)	0.724
Impella CP	1 (8%)	4 (25%)	0.624
Impella 5.0	1 (8%)	6 (32%)	0.202
extra corporeal membrane oxygenation	8 (67%)	2 (11%)	0.002 *
mean arterial pressure (mmHg)	73 (61, 77)	73 (63, 76)	0.880
pulse pressure (mmHg)	31 (9, 51)	32 (8, 56)	0.745
hemoglobin (g/dL)	10.9 (8.8, 11.9)	10.2 (9.3, 10.9)	0.516
platelet (× 10^4^/μL)	7.4 (4.1, 11.2)	8.9 (5.9, 14.1)	0.264
total bilirubin (mg/dL)	1.8 (1.1, 2.8)	1.3 (0.8, 2.2)	0.247
serum creatinine (mg/dL)	1.24 (1.02, 1.72)	1.32 (0.78, 1.81)	0.792
C-reactive protein (mg/dL)	6.9 (1.0, 20.0)	4.2 (3.1, 11.8)	0.655
B-type natriuretic peptide (pg/mL)	458 (222, 817)	402 (184, 617)	0.715
lactate dehydrogenase (U/mL)	1297 (900, 2247)	633 (265, 1389)	0.007 *
activated partial thromboplastin time (s)	53.1 (37.9, 65.4)	50.1 (43.5, 67.7)	0.871
antiplatelets	8 (67%)	8 (42%)	0.273
support duration (days)	6.5 (3, 8)	3 (2, 6)	0.149

* *p* < 0.05 by Mann–Whitney U test or Fisher’s exact test, as appropriate.

## Data Availability

Data are available by the corresponding author upon reasonable requests.

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
