# Peer review of "Impact of the Severity of Acquired von Willebrand Syndrome on the Short-Term Prognosis in Patients with Temporary Mechanical Circulatory Support"

_medicina, 2022, doi:10.3390/medicina58020238_

Round 1

Reviewer 1 Report

In the manuscript medicina-1562351, authors present a study in which the von Willebrand factor multimer analyses were performed to a pool of selected patients who received temporary mechanical circulatory support from 2018 to 2021.  

The article is well written, and the outcome is clear to follow, and the content of the study is relevant for publication in Medicina. I have a minor suggestion for the figure description of the figure 01, It would be better if authors could label panels of the figure separately and descriptively write the figure title. That would help readers to follow the figure easily.

Author Response

General comment

In the manuscript medicina-1562351, authors present a study in which the von Willebrand factor multimer analyses were performed to a pool of selected patients who received temporary mechanical circulatory support from 2018 to 2021. 

Response

We sincerely express our great appreciation for the reviewer’s recommendation to our manuscript. According to the reviewer’s comment, we revised our manuscript. Please read through our revised manuscript.

 Comment 1

The article is well written, and the outcome is clear to follow, and the content of the study is relevant for publication in Medicina. I have a minor suggestion for the figure description of the figure 01, It would be better if authors could label panels of the figure separately and descriptively write the figure title. That would help readers to follow the figure easily.

Response 1

Thank you for the reviewer’s useful comment. In Figure 1, vWF multimers were divided into three parts (large, medium, and small and smallest multimers) according to their gravities (A). The large multimer ratio was calculated as a ratio of the large multimer area to the total multimers area (B). The large multimer index was defined as a large multimer ratio normalized by the value of control (C). We revised the Figure 1 as suggested by the reviewer.

Before 1

Figure 1

See previous Figure 1

Methods

Measurement of vWF multimer index:

vWF large multimers were quantified from … in the next lane of the same vWF multimer analysis (Figure 1), referencing the previous studies [8, 9].

Figure legends

Figure 1: The vWF large multimer index was defined as the ratio of the large multimer area of a patient to that of a control whose plasma was analyzed in the next lane of the same vWF multimer analysis

vWF, von Willebrand factor.

After 1

Figure 1

 See updated Figure 1

Methods

Measurement of vWF multimer index:

vWF large multimers were quantified from images of vWF multimer analysis by gel electrophoresis (Figure 1A). Using ImageJ, the central one-third of the lane was scanned. vWF multimers were divided into three parts (large, medium, and small/smallest multimers). A large multimer ratio was calculated as the ratio of the large multimer area to the total multimers area (Figure 1B). A vWF large multimer index was defined as a large multimer ratio normalized by the control one (Figure 1C).

 Figure legends

Figure 1: vWF multimer analysis by gel electrophoresis (A), the calculation of large multimer ratio (B), and the calculation of large multimer index (C).

vWF multimers were divided into three parts (large, medium, and small and smallest multimers) according to their gravities (A). The large multimer ratio was calculated as a ratio of the large multimer area to the total multimers area (B). The large multimer index was defined as a large multimer ratio normalized by the value of control (C).

Reviewer 2 Report

Overall, this is an interesting clinical study that contains a classical evaluation of predictors based on logistic regression analysis.

  1. Figure 3. Please specify this is a receiver operating characteristic (ROC) curve. X-axis corresponds to false positive rate or 1-specificity and Y-axis to true positive rate or sensitivity. Please specify that AUC is the C-statistic. Please also indicate the 95% confidence interval of the C-statistic. This is essential to interpret the C-statistic.
  2. Why do you define a cut-off value? How is it defined? Any choice has an effect on the sensitivity and the specificity. In other words, the ideal cut-off value depends on the objective of the test.
  3. Table 2 is very instructive. How are the odds values expressed? Per standard deviation increase or per unit increase. This information is essential to interpret odds ratios.
  4. A Table with selected univariable predictors and corresponding C-statistics and 95% confidence intervals would be useful. C-statistics can be easily interpreted.
  5. I doubt whether a multivariable model makes a lot of sense. There is the problem of overfitting. For obvious reasons, the number of patients in the study is limited. A rule of thumb is that 10 patients are required per predictor. More specifically, the rule states that one predictive variable can be studied for every ten events. For logistic regression the number of events is given by the size of the smallest of the outcome categories (here the surviving group or non-surviving group). The non-surviving group consists of only 8 patients…
  6. With regard to a theoretical multivariable model, it should be evaluated whether or not a covariable adds significant information (chi-square test).
  7. The correct terminology is univariable and multivariable and not univariate and multivariate.

 Author Response

General comment

Overall, this is an interesting clinical study that contains a classical evaluation of predictors based on logistic regression analysis.

Response

We sincerely express our great appreciation for your comments that have improved our manuscript significantly. We tried our best to answer all your comments and revised our manuscript. Please read through our responses and updated manuscript.

Comment 1

Figure 3. Please specify this is a receiver operating characteristic (ROC) curve. X-axis corresponds to false positive rate or 1-specificity and Y-axis to true positive rate or sensitivity. Please specify that AUC is the C-statistic. Please also indicate the 95% confidence interval of the C-statistic. This is essential to interpret the C-statistic.

Response 1

Figure 3 displays the ROC curve with X-axis corresponding 1-specificity and Y-axis corresponding sensitivity. AUC indicates C-statistic. 95% confidence interval of the C-statistic was between 0.614 and 0.995.

Before 1

Methods

None

After 1

Methods

The area under curve was assumed as C-statistic.

Comment 2

Why do you define a cut-off value? How is it defined? Any choice has an effect on the sensitivity and the specificity. In other words, the ideal cut-off value depends on the objective of the test.

Response 2

A cutoff was derived by applying Youden method in the ROC analysis. We updated the methods section.

Before 2

Methods

A cutoff of vWF large multimer index for 30-day survival was calculated by the receiver operating characteristics analysis and the cohort was stratified into two groups: lower vWF multimer index group and not lower group.

After 2

Methods

A cutoff of vWF large multimer index for 30-day survival was calculated by Youden method in the receiver operating characteristics analysis and the cohort was stratified into two groups: lower vWF multimer index group and not lower group.

Comment 3

Why Table 2 is very instructive. How are the odds values expressed? Per standard deviation increase or per unit increase. This information is essential to interpret odds ratios.

Response 3

We appreciate the important questions. The odds values are expressed per unit increase in continuous variables. We added the description to the manuscript.

Before 3

Methods

Statistical assessments:

The impacts of clinical variables including vWF large multimer index on the primary and secondary endpoints were investigated by the logistic regression analyses.

After 3

Methods

Statistical assessments:

The impacts of clinical variables including vWF large multimer index on the primary and secondary endpoints were investigated by the logistic regression analyses. The odds values are expressed per unit increase in continuous variables.

Comment 4

Why A Table with selected univariable predictors and corresponding C-statistics and 95% confidence intervals would be useful. C-statistics can be easily interpreted.

Response 4

We stated C-statistics and 95% confidence intervals of all three variables that were enrolled in the multivariable analysis in Table 3. Please see a newly added Table 3.

Before 4

Results

None

After 4

Results

Impact of vWF large multimer index on the primary endpoint:

C-statistics of these three variables that were enrolled in the multivariable analysis were stated in Table 3.

See Table 3

Comment 5

Why I doubt whether a multivariable model makes a lot of sense. There is the problem of overfitting. For obvious reasons, the number of patients in the study is limited. A rule of thumb is that 10 patients are required per predictor. More specifically, the rule states that one predictive variable can be studied for every ten events. For logistic regression the number of events is given by the size of the smallest of the outcome categories (here the surviving group or non-surviving group). The non-surviving group consists of only 8 patients…

Response 5

Yes. We completely agree with the reviewer. Given a small number of the events, it is challenging to enroll many variables in the multivariable analysis. On the contrary, multivariable analysis is of great importance to adjust for potential confounders that might also have considerable prognostic impacts. We should be cautious to interpret the finding of our multivariable analysis. We strengthened the limitation section.

Before 5

Discussion

Limitations:

We performed multivariate analyses using only three variables given a relatively small cohort size. Other uninvestigated confounders might have existed.

After 5

Discussion

Limitations:

Given a small event number, it is challenging to enroll multiple variables in the multivariable analyses. We enrolled three variables in this study, but this has a potential of overfitting. We should pay caution to interpret the finding of multivariable analysis.

Comment 6

With regard to a theoretical multivariable model, it should be evaluated whether or not a covariable adds significant information (chi-square test).

Response 6

We appreciate the important questions. We added the prognostic impact of potential confounders that were enrolled in the multivariable analysis.

Deceased (N=8)

Alive (N=23)

p value

Platelet < 9.8 ×104 /μL

7 (88%)

11 (48%)

0.095

vWF large multimer index <59.9(%)

7 (88%)

5 (22%)

0.002

Extracorporeal membrane oxygenation

6 (75%)

4 (17%)

0.006

Before 6

Results

None

After 6

Results

Impact of vWF large multimer index on the primary endpoint:

Both potential confounders also tended to be association with 30-day survival (48% versus 88%, p=0.095, 17% versus 75%, p=0.006, respectively).

Comment 7

The correct terminology is univariable and multivariable and not univariate and multivariate.

Response 7

We apologize for the mistakes of the terminology. We corrected the word as the term as pointed out.

 Before 7

Methods

Statistical assessments:

Variables with p <0.05 in the univariate analyses were included in the multivariate analyses.

See previous Table 2

 After 7

Methods

Statistical assessments:

Variables with p <0.05 in the univariable analyses were included in the multivariable analyses.

Table 2

See updated Table 2

Round 2

Reviewer 2 Report

The authors have provided an ad rem response to the comments of the reviewer. The revised manuscript reflects an adequate effort to address my comments.